# Anionic Phospholipids Shift the Conformational Equilibrium of the Selectivity Filter in the KcsA Channel to the Conductive Conformation: Predicted Consequences on Inactivation

**DOI:** 10.3390/biomedicines11051376

**Published:** 2023-05-05

**Authors:** María Lourdes Renart, Ana Marcela Giudici, Carlos Coll-Díez, José M. González-Ros, José A. Poveda

**Affiliations:** IDiBE—Instituto de Investigación, Desarrollo e Innovación en Biotecnología Sanitaria de Elche, Universidad Miguel Hernández, 03202 Elche, Spain; lrenart@umh.es (M.L.R.); marcela@umh.es (A.M.G.); ccoll@umh.es (C.C.-D.)

**Keywords:** potassium channels, C-type inactivation, selectivity filter conformation, protein thermal stability, fluorescence anisotropy, ion-protein interactions, lipid modulation

## Abstract

Here, we report an allosteric effect of an anionic phospholipid on a model K^+^ channel, KcsA. The anionic lipid in mixed detergent–lipid micelles specifically induces a change in the conformational equilibrium of the channel selectivity filter (SF) only when the channel inner gate is in the open state. Such change consists of increasing the affinity of the channel for K^+^, stabilizing a conductive-like form by maintaining a high ion occupancy in the SF. The process is highly specific in several aspects: First, lipid modifies the binding of K^+^, but not that of Na^+^, which remains unperturbed, ruling out a merely electrostatic phenomenon of cation attraction. Second, no lipid effects are observed when a zwitterionic lipid, instead of an anionic one, is present in the micelles. Lastly, the effects of the anionic lipid are only observed at pH 4.0, when the inner gate of KcsA is open. Moreover, the effect of the anionic lipid on K^+^ binding to the open channel closely emulates the K^+^ binding behaviour of the non-inactivating E71A and R64A mutant proteins. This suggests that the observed increase in K^+^ affinity caused by the bound anionic lipid should result in protecting the channel against inactivation.

## 1. Introduction

The more we know about biological membranes, the more we realize the variety of ways by which lipids interact with membrane proteins to influence their structure and function, thus modulating the overall cellular response. Such influence is sometimes exerted through an indirect effect on the protein by altering physical properties of the membrane bilayer, such as tension, curvature, lateral pressure, thickness, hydrophobic mismatch, or others [1,2,3]. In some other instances, however, there is a direct effect caused by lipid binding to specific sites on the protein [4,5,6]. In this paper, we use the KcsA channel in mixed detergent–lipid micelles (DLMs) as an experimental system in which to study the modulation of this model membrane protein by lipids. Although departing from a classical liposome-like membrane bilayer model, DLMs have the advantage of contributing much less scattering in spectroscopic (fluorescence) measurements due to their small size. Moreover, because DLMs lack an organized bilayer, the observations of lipid effects on the proteins should mainly reflect the direct interactions of the lipids, acting truly as specific protein effectors.

KcsA is a prokaryotic K^+^ channel from *Streptomyces lividans*. By X-ray crystallography, the structure of this channel has been determined, showing a homotetrameric transmembrane protein. Two transmembrane segments (TM1 and TM2) in the form of alpha-helices, as well as the cytoplasmic N- and C-termini, stand out in each monomer (Figure 1). In the case of the four C-termini, these form a bundle of helices that acts as a channel gate (the inner gate) that is sensitive to pH [7]. The channel pore consists of an aqueous cavity flanked by a short tilted helix (pore helix) as well as the selectivity filter (SF), whose TVGYG sequence is homologous to that of eukaryotic K^+^ channels, acting as a secondary gate (the outer gate). It contains four binding sites for K^+^, formed by the backbone carbonyls of the SF residues (sites S1–S4 of the crystal, from the extracellular to the intracellular side) [8]. This SF is able to adopt two different conformations at high and low K^+^ concentration [8,9,10,11,12,13,14,15,16,17,18]. For the second one, ions would bind only to sites S1 and S4, so there would be no ions in the center (sites S2 and S3), resulting in a collapsed non-conducting conformation. At high K^+^ concentration, a conformational change would occur, in which the ions would access all the SF sites, enabling conduction [8,12]. In the case of the non-permeant Na^+^ ion, it is not able to induce such a conformational change, remaining in a situation similar to that of K^+^ at low concentrations, i.e., with the SF collapsed and with the internal sites S2 and S3 empty [19].

Ion binding to the SF of KcsA in detergent micelles has also been studied through different assays based on the cation-dependent variation of (i) the intrinsic fluorescence emission spectrum, (ii) the steady-state anisotropy, or (iii) the fluorescence monitoring of thermal denaturation of the protein [23,24,25,26]. In these studies, the published crystal structures of the channel protein were always used as the basis to interpret the binding results. On the one hand, the addition of K^+^ induced a blue shift in the fluorescence emission spectrum of the protein, together with an increase in its intensity, in any case greater than those caused by the addition of Na^+^. This is an expected effect since K^+^ induces a greater conformational change than Na^+^ in the SF of KcsA, according to crystallographic data [23]. On the other hand, the use of tryptophan to phenylalanine mutants of KcsA allowed us to assign the cation-induced changes in fluorescence mainly to residues Trp67 and Trp68 [26,27]. These residues are located in the short helix of the protein with their side chains practically in contact with the polypeptide backbone of the selectivity filter [11,28,29]. From these results, it can be concluded that, as indicated by previous X-ray studies, it is the selectivity filter and/or its closest surroundings that are involved both in the binding of cations and in the conformational changes from which this binding originates. Finally, another experimental observable with which ion binding to KcsA can be monitored is the thermal denaturation of the protein, followed by its intrinsic fluorescence; the apparent *t_m_* (midpoint denaturation temperature in degrees Celsius) of the thermal transition from the native to the denatured state can be calculated, and its major advantage being the wide range of variation of this parameter [24,26]. With these experiments, it was detected that for non-permeant Na^+^ or Li^+^ ions, there would only be a low affinity binding event [25], however, in the case of permeant K^+^, Rb^+^, Tl^+^, and even Cs^+^ ions, two binding events were detected as their concentration increases—one high and one low affinity [25]. By comparing these results with previous ones obtained by crystallography, we were able to hypothesize that the first binding event for any of these ions would correspond to the binding to the S1 and S4 sites of the SF as defined by crystallography, giving rise to a non-conducting collapsed conformation. The higher affinity shown by the permeant ions (micromolar *K_D_* values) would ensure the displacement of competing non-permeant cations (millimolar *K_D_*) and, thus, a selectivity towards these ions. The second binding event of lower affinity (millimolar *K_D_* values) would favour the dissociation of the ions and, thus, their permeation. In this case, as described by crystallography, only permeant ions would be able to induce this change of state and occupy on average all the S1–S4 sites of the SF, thus facilitating their flux [13]. Derived from these data, it is interesting to note that the differences in affinity between permeant and non-permeant cations, as well as the similarities in ion binding within each group, correlate with their relative permeabilities to K^+^, which allows us to conclude that ion channel selectivity is strongly determined by the binding of cations to the channel [25]. Additional evidence to support the role of the SF as the main site for cation binding comes from the altered binding observed in channel mutants affecting the SF such as M96V or E71A [24,30], or using specific SF channel blockers [24,31].

The activity cycle of KcsA has been defined in four stages with allosteric communication between the two gates of the channel being a key factor. Thus, at neutral pH and in the presence of K^+^, the closed/conductive state would be the predominant one, in which the cytoplasmic helical bundle (inner gate) would prevent the flow of ions, leaving the SF (outer gate) in a conductive conformation. Upon lowering the pH, it would switch to an “open/conductive” state since the internal gate would open, making KcsA a proton-activated channel [32,33,34]; however, this conformation is not stable and drifts within a few seconds to one in which the outer gate closes, preventing ions from permeating the channel, thus defining an “open/inactivated” state [35,36]. This process is modulated by a network of interactions, among which include the E71, D80, and W67 residues of each subunit located behind and in contact with the SF and known as the inactivation triad, stand out [11,37,38,39]. When the pH is back to neutral, the cycle is completed by closing the internal gate and returning the SF to its resting conductive state [40].

The examination of the KcsA crystal structure reveals that it contains noncovalently bound lipid [8,41], initially modeled as diacylglycerol and, later on, identified as phosphatidylglycerol (PG) [41,42]. The crystallographic evidence and other studies [43] conclude that the PG binding sites on the KcsA protein have the features of so-called “non-annular” lipid binding sites on the protein surface [44] between the TM1 and TM2 of each pair of adjacent KcsA monomers. The negatively charged polar headgroup of the lipid interacts with nearby positively charged R64 or R89 residues in a not-too-specific manner, as there are only minor differences in the binding affinities of different negatively charged phospholipids [43,45,46]. In addition to the R64 and R89 residues, other amino acids were described to be involved in the binding of anionic lipids to the “non-annular” sites in KcsA, providing the required architecture to hold the lipid molecule in the crevices between subunits [47]. Similar “non-annular” sites containing bound lipids have been documented in several other membrane proteins where they also play important roles in modulating protein features ranging from their correct folding to their optimal activity [2,5,48]. In KcsA in particular, the binding of anionic phospholipids to such sites are crucial for the correct insertion of the tetrameric KcsA channel into the cell membrane and are needed to completely refold the channel upon trifluoroethanol unfolding [49,50]. Anionic phospholipids also increase the thermal stability of the KcsA protein much more efficiently than zwitterionic phospholipids [47] by holding together adjacent KcsA subunits through protein–lipid interaction at the intersubunit, “non-annular” lipid binding sites [4,5,20,47,51]. Finally, anionic phospholipids are required for optimal activity of the KcsA channel [52,53]. A reduction in the macroscopic rate of inactivation and an increase in the single-channel mean open time are also observed as the concentration of anionic phospholipids is increased; these effects are attributed to specific lipid interaction with the “non-annular” arginines [20]. Such interaction and its role on inactivation has been documented by functional, structural and molecular dynamics simulation studies [20,21,22,54]. Still, the molecular mechanisms involved in the alteration of the selectivity filter by lipid binding need to be further addressed. Additionally, anionic phospholipids have been reported to interact with Arg residues at the N-terminal domain, where they could modulate the opening of the channel’s inner gate [53].

Our main goal in this paper is to study the effects of lipids in modulating the interaction between the channel and relevant permeating and non-permeating cations. Monitoring such interactions by the thermal denaturation or fluorescence anisotropy of the detergent-solubilized KcsA protein have proven to be a very useful tool to illustrate the conformational equilibrium at the channel’s SF throughout its functional cycle.

## 2. Materials and Methods

### 2.1. Materials

N-dodecyl-β-D-maltoside (DDM) ULTROL^®^ Grade was from Merck. Hepes, succinic acid, N-methyl-D-glucamine (NMG), NaCl, and KCl, were from Sigma-Aldrich. Ni^2+^-Sepharose Fast Flow Resin was from GE Healthcare. 1-palmitoyl-2-oleoyl-glycero-3-phosphocholine (POPC) and 1-palmitoyl-2-oleoyl-sn-glycero-3-phosphate (sodium salt) (POPA) were from Avanti Polar Lipids. The stocks of POPC and POPA were usually made in chloroform:methanol (3:1) at 5–10 mg/mL, and appropriate aliquots were then evaporated under vacuum for at least 3 h. The lipids where then solubilized in the samples buffer under gentle agitation for at least 3 h.

### 2.2. KcsA Heterologous Expression and Purification

The heterologous expression of KcsA wild-type (WT) and the mutants W26,68,87,113F (W67); W26,68,87,113F E71A (E71A W67); W26,68,87,113F R64A (R64A W67); and L90C were performed in *E. coli* M15 (pRep4) and purified by Ni^2+^/His-tag affinity chromatography according to previous reports [27,49]. The tetrameric state of the protein was routinely checked by SDS-PAGE (12%) [55].

### 2.3. Fluorescence Monitoring of Cation Binding through Thermal Denaturation

Thermal denaturation of WT KcsA channel was performed as described previously [26] using a Varian Cary Eclipse spectrofluorometer to record the temperature dependence of the protein intrinsic emission fluorescence at 340 nm after excitation at 280 nm. In these experiments, the protein was diluted to 1 µM concentration (in terms of monomers of KcsA) in either 20 mM Hepes buffer, pH 7.0, containing 5 mM DDM and 1.5 mM NaCl (pH 7.0 buffer) or 10 mM succinic acid buffer, pH 4.0, containing 5 mM DDM and 1.5 mM NaCl (pH 4.0 buffer). To prepare the DLMs, the above samples were supplemented with 0.5 mM of the desired phospholipid. For the cation titrations, aliquots from stock solutions of either NaCl or KCl were added to the samples prior to the thermal denaturation recordings to provide the desired final cation concentrations.

The midpoint temperature of dissociation and unfolding of the tetramer (*t_m_*, in Celsius) was calculated from the thermal denaturation curves by fitting a two-state unfolding model to the data [47]. The dissociation constants of the KcsA-cation complexes (*K_D_*) can be estimated from:(1)$$\frac{{\Delta T}_{m}}{T_{m}}=\frac{T_{m}-{\left(T_{m}\right)}_{0}}{T_{m}}=\frac{R({T_{m})}_{0}}{{\Delta H}_{0}}ln\left(1+\frac{\left[L\right]}{K_{D}}\right)$$
where *T_m_* and (*T_m_*)_0_ refer to the denaturation temperature (in Kelvin) for the protein in the presence and absence of ligand, respectively, *R* is the gas constant, and Δ*H*_0_ is the enthalpy change upon protein denaturation in the absence of ligand.

### 2.4. Steady-State and Time-Resolved Fluorescence Measurements

Spectroscopic measurements in the DLMs at different cation concentrations, were carried out exactly as described above for the thermal denaturation assays, except that a 5–6 µM (monomer based) final protein concentration was used throughout.

Steady-state fluorescence measurements were performed on a Horiba Jobin Yvon Fluorolog3 (Piscataway, NJ, USA) or SLM-8000 C (SLM Aminco, Urbana, IL, USA) spectrofluorometer, using 0.5 cm pathlength quartz cuvettes at room temperature. Steady-state anisotropy was measured at 340 nm using an excitation wavelength of 300 nm and calculated as:(2)$$<r>=\frac{I_{VV}-G\times I_{VH}}{I_{VV}+2\times GI_{VH}}$$
where *I_VV_* and *I_VH_* are the fluorescence intensities (blank subtracted) of the vertically and horizontally polarized emission when the sample is excited with vertically polarized light, respectively. The G-factor corresponds to the instrument correction factor (*G* = *I_HV_*/*I_HH_*). Ten measurements were done for each sample [27].

Time-resolved fluorescence and anisotropy measurements with picosecond resolution were obtained using the time-correlated single-photon timing (SPT) technique as in [27].

The W67–W67 intersubunit distances were calculated from the anisotropy decays of W67 KcsA or E71A W67 KcsA and R64A W67 KcsA mutant channels in either DLMs or in plain detergent protein micelles, respectively, as described previously [27,31].

### 2.5. Calculation of the K^+^ Binding Affinity to KcsA from Changes in the Steady-State Anisotropy Values

When monitored by the changes in steady-state anisotropy (<*r*>), the binding of K^+^ to W67 KcsA presented a sigmoidal behavior that was used to calculate the binding affinity of the channel for this permeant cation, as is in [31]. Briefly, the *K_D_* values can be calculated from the following equation:(3)$$<r>={<r>}_{NC}+(Q{<r>}_{C/I}-{<r>}_{NC})\frac{{{[X}^{+}]}^{h}}{K_{D}^{h}+{{[X}^{+}]}^{h}}$$
where <*r*>*_NC_* and <*r*>*_C/I_* are the steady-state anisotropy of the initial (non-conductive, *NC*) and final (conductive or inactivated, *C*/*I*) states, respectively, [*X*^+^] is the concentration of the cation, *K_D_* is the dissociation constant, and h is the Hill coefficient. The *Q* parameter represents the relative change between the quantum yield of the non-conductive and conductive (or inactivated) forms.

### 2.6. DLS Measurements

Plain and mixed micelle size determination was carried out by dynamic light scattering (DLS) technique using a Malvern Zetasizer Nano-ZS instrument (Worcestershire, UK) with a Helium–Neon laser (λ = 633 nm). Measurements were performed at 25 °C in triplicate at an angle of 173°.

### 2.7. Statistics

Statistical analysis was performed using Graphpad Prism version 5 (Graphpad Software). The statistical significance of observed differences was assessed by two-tailed unpaired *t*-student test.

## 3. Results

### 3.1. Thermal Denaturation of WT-KcsA in Mixed Detergent–Lipid Micelles (DLMs)

Our purpose in this paper is to study the possible changes taking place in the cation-dependent, conformational equilibrium of the KcsA channel upon lipid binding, as a way to explain their modulatory role. To do so, mixed detergent–lipid micelles (DLMs) have been used, which minimizes light scattering in our spectroscopic studies due to their small size relative to lipid membranes. In addition, the lack of a membrane bilayer structure in the DLMs allows us to attribute the observed effects to lipid-channel binding instead of to changes in bilayer properties. In this respect, we reported previously that lipids binding to the channel in DLMs could be easily illustrated by monitoring the thermal denaturation of the KcsA protein in the presence of increasing concentrations of different phospholipids within the micelles [47]. The mid-point protein denaturation temperature (*t_m_*) increases considerably, even at lipid concentrations as low as 10^−4^–10^−5^ M, much more markedly for anionic phospholipids over zwitterionic ones, thus, providing an excellent experimental observable with a wide range of variation [47]. Such lipid-induced stabilization is susceptible to further rise in the presence of increasing concentrations of cations, as shown below. For the above reasons, we decided to do cation titrations in thermal denaturation experiments using the WT KcsA channel in DLMs containing either POPA or POPC and at the indicated pHs (see Section 2). Figure 1 and Figure 2 illustrate such titrations using Na^+^ or K^+^ as non-permeating or permeating cation species, respectively.

In the Na^+^ titrations (Figure 1A,B), besides the initial thermal stabilization of the KcsA protein induced by the lipids in the DLMs, the responses to increasing Na^+^ concentration are somewhat similar to that observed in the plain detergent micelles. The analysis of the thermal denaturation patterns, were carried out as described in detail earlier [25,26], with supplementary information available in [24]. Single binding events are observed in all cases and well described by the binding curves throughout the Na^+^ concentration range, which allow us to calculate the apparent dissociation constants (*K_D_*) for Na^+^ binding to KcsA (Table 1). Regardless of whether we started with the channel at pH 7.0 or 4.0, with a closed or an open inner gate, respectively, there is a similarity in the estimated *K_D_* for the observed low-affinity, single Na^+^ binding event. This event has been related to the adoption of a collapsed conformation of the KcsA SF [8,23,26,56], which seems not to be sensitive to the presence or to the absence of lipids in the DLMs.

Figure 2 shows K^+^ titration patterns in the DLMs, which are also reminiscent of those reported previously in plain detergent micelles, including the larger cation-induced thermal stabilization of the KcsA protein, compared to that caused by Na^+^, or the occurrence of two successive binding events as the K^+^ concentration is increased [25,26,57]. As concluded previously, the first high-affinity K^+^ binding event is associated with the stabilization of the non-conductive, collapsed SF conformation (S1 and S4 sites occupied), whereas the second low-affinity set correspond to the self-induced conformational change to a conductive, fully occupied (sites S1 to S4, Figure 1) SF form [8,12,26]. In previous reports using plain detergent–protein micelles, the analysis of thermal denaturation at increasing K^+^ concentration allows for the determination of the dissociation constants for both binding events [24,25,26]. In this case, however, the cumulative thermal stabilization caused by lipids in the DLMs, particularly POPA, added to that from the increasing K^+^ concentrations, causes the *t_m_*’s determination above a certain K^+^ concentration to exceed the capacity of our thermostating system (*t_m_* values near or over 100 °C). Therefore, the resulting binding curves only cover a limited range of K^+^ concentrations, from which *K_D_* determinations become uncertain (Table 1), and thus, a precise analysis of the data is prevented. Still, the data can be analyzed taking into account these limitations to conclude that POPA, in particular, and at pH 4.0 (Figure 2B) shifts the titration curve to lower K^+^ concentrations, that is, increasing the affinity of the channel to bind the cation, especially in the second binding event (Table 1). Such effects can perhaps be more clearly observed in the semilog plots included as inserts in the panels. Although just an approximation, this could be a relevant observation since channel inactivation requires emptying the SF of cations [58,59,60], which would not occur as easily if the affinity of the SF for K^+^ is increased. In other words, the presence of POPA in the DLMs could be protecting the channel against inactivation. This could explain why much lower inactivation rates are observed in patch-clamp measurements of KcsA in reconstituted membranes containing the anionic phospholipid [20].

### 3.2. Binding of K^+^ to the W67-KcsA Mutant in DLMs Followed by Fluorescence Anisotropy

Because of the limitations of the thermal denaturation assay to assess the binding of K^+^ to the KcsA channel in the DLMs with accuracy, as indicated above, we turned to use fluorescence anisotropy measurements, which are taken at room temperature. For this we used the W67 KcsA mutant (W26,68,87,113F KcsA mutant), which only has a single tryptophan residue left (W67) at the channel pore helix, just behind the SF. This is a region sensitive to the presence of K^+^ within the SF, particularly at the S2 and S3 sites, which are heavily involved in channel inactivation [27,61,62,63]. Previous reports from our group have shown that W67 KcsA is structurally and functionally similar to WT KcsA [27]. Still, we started by optimizing the experimental conditions to form the DLMs. Dynamic light scattering measurements of DLMs made from W67 KcsA in detergent and variable concentrations of phospholipids (see Section 2) showed that, below 0.5 mM of either POPA or POPC, the size of the resulting DLMs did not increase significantly with respect to plain detergent–protein micelles (10–12 nm average diameter). Furthermore, thermal denaturation experiments at increasing lipid concentrations, similar to those reported previously [47], showed that 0.5 mM of either lipid in the DLMs already causes a maximum effect on increasing the thermal stability of the protein. Therefore, as is in the above DLMs from WT KcsA, we chose 0.5 mM lipid in the DLMs to secure both a maximum saturated effect of the lipid on the protein and a minimum size of the resulting DLMs to minimize light scattering contributions to the fluorescence measurements.

In spite of the advantages of the anisotropy monitoring mentioned above, it should be noticed that changes in <*r*>*ss* in the W67 KcsA are a consequence of the changes in the homo-FRET rates among the W67 residues, i.e., changes in the W67–W67 intersubunit distances [27]. In plain detergent–protein micelles and at pH 7.0, we reported previously that the single binding event of Na^+^, as well as the first binding event of K^+^ at low concentration (the first equilibrium from an empty SF to a collapsed/non-conductive conformation), are associated with only a small modification in the intersubunit distances, and therefore, it is expected that anisotropy changes under those conditions provide low sensitivity to estimate the *K_D_* value. Indeed, we observed here that increasing the concentration of Na^+^ in the samples has little effect on the anisotropy measurements, particularly at pH 4.0. This suggests that Na^+^ binding to W67 KcsA does not significantly affect the conformation of the collapsed protein around the short helix where the W67 reporter groups are located. For this reason, and also because Na^+^ binding was adequately described by thermal denaturation, Na^+^ titrations of the DLMs were not further pursued in this work. Likewise, according to the expectations, changes in the anisotropy observed at low K^+^ concentrations, corresponding to the first K^+^ binding event by the channel, are also minor, and therefore, in K^+^ titrations such as those shown in Figure 3, we paid attention particularly to the K^+^ concentration range covering the second K^+^ binding event to W67 KcsA in the DLMs. This second event is associated with the filling of the innermost binding sites of the SF (namely S2 and S3 binding sites) and in plain detergent–protein micelles at pH 7.0, causing nearly a 3 Å modification in the intersubunit distances [27]. Fortunately, this is precisely the K^+^ concentration range that could not be properly addressed from the above thermal denaturation experiments, and it is most relevant in ion conduction and channel inactivation.

Both panels in Figure 3 show that, regardless of either pH or the presence of a given phospholipid in the DLMs, the W67 KcsA channel seemingly transits in a K^+^ concentration-dependent manner between two states, characterized by steady-state anisotropy values approaching 0.165 and near 0.130, respectively. We reported previously that the square geometry of the four W67 residues within the channel, gives raise to W67–W67 homo-FRET, from which intersubunit distances can be estimated from time-dependent fluorescence anisotropy decays [27]. Moreover, such experimentally determined intersubunit distances have an excellent linear correlation with the steady-state anisotropy values (Figure 4E in [31]). Therefore, the steady-state anisotropies given in Figure 3 serve to predict an approximately 18 and 15 Å intersubunit distance, respectively, for the collapsed and the resting conductive states of the partly empty and fully occupied SF. The above intersubunit distances are formally correct only for the plain detergent–protein micelles, in which such linear correlation was experimentally established [31]. Nonetheless, the similarities between the steady-state anisotropies in the plain detergent–protein micelles and either one of the DLMs predicts a similar situation in the latter, suggesting that the initial and final states of the channel’s SF are not significantly modified by the presence of either phospholipid in the DLMs. Following the same procedures reported earlier [27], this expectation was indeed validated by measuring time-dependent fluorescence intensity and anisotropy decays, along with a determination of the rotational relaxation time in the DLMs, to experimentally determine the intersubunit distances at saturating (200 mM) K^+^ concentrations. The intersubunit distances determined for either of the DLMs under those conditions, were almost identical to those determined previously for W67 KcsA in plain detergent–protein micelles [27].

Figure 3A shows that at pH 7.0, where the inner gate is closed and the SF is being occupied by K^+^ and converted into the conductive state, the affinities of K^+^ to bind to the channel are comparable in the plain detergent–protein micelles and in either one of the DLMs, suggesting that none of the phospholipids tested significantly modify the binding of K^+^ to the channel. The apparent *K_D_* for the binding processes were estimated from the anisotropy vs. K^+^ concentration curves, as described in the Section 2, and are given in Table 2. Likewise, at pH 4.0, when the inner gate is open (Figure 3B), K^+^ binding is also practically identical in plain detergent–protein micelles and in POPC-containing DLMs. Conversely, however, K^+^ binding differs significantly from the above in DLMs containing POPA (Figure 3B), in which the affinity of K^+^ to bind the channel clearly increases (Table 2). This conclusion is similar to that reached above from the qualitative interpretation of thermal denaturation experiments (Figure 2B). This further suggests that POPA acts specifically to increase the affinity for K^+^ binding when the inner gate of the channel is open, thereby partly preventing the SF from emptying of bound K^+^, which is a prerequisite for channel inactivation to occurs.

### 3.3. Binding of K^+^ to Model Non-Inactivating W67 KcsA Mutants Followed by Fluorescence Anisotropy

In an attempt to confirm that the above effects of POPA in preventing the inactivated state in W67 KcsA in DLMs resembles the behavior of well-known non-inactivating channel mutants, we prepared E71A W67 and R64A W67 KcsA mutants and did K^+^ titration experiments followed by fluorescence steady-state anisotropy. The E71A mutant was first described by Cordero-Morales and collaborators [11]; it lacks a complete “inactivation triad” (E71-W67-D80) because of the absence of the E71 residue and presents a much higher open probability and mean open time than the WT KcsA channel. On the other hand, the R64A mutant, lacking one of the two R64 and R89 “non-annular” arginine residues involved in anionic phospholipid binding, was also described first by Cordero-Morales and collaborators [11], and likewise, it presents a higher open probability and mean open time than the WT channel, although slightly lower than the E71A mutant counterpart. Interestingly, the addition of anionic phospholipids to these mutant channels reconstituted in asolectin giant liposomes did not modify their macroscopic current profile in electrophysiological measurements [20,64]. Furthermore, MD simulation studies highlighted the main role of the R64 residue not only in mediating the specific binding of anionic lipids to the “non-annular” binding sites, but also in translating such binding to specific conformational changes in the SF [20,54].

Figure 4A,B show that both of these non-inactivating mutant channels present a similar trend when binding to K^+^ almost independently of the conformational state of the channel’s inner gate reached at pH 7.0 or 4.0. The estimated *K_D_* values all fall in the low mM range (Table 2), which are very similar to those determined for the W67 KcsA sample in the POPA-containing DLMs at pH 4.0.

The comparison between the non-inactivating effects of POPA on the W67 KcsA and the behavior of the non-inactivating model channels can be more clearly seen in Figure 5 and reinforces the idea that POPA protects the channel against inactivation, specifically at pH 4.0 by increasing K^+^ affinity. On the contrary, the behavior of the channel in POPC DLMs does not differ from that in plain DDM micelles, showing a clear affinity loss for K^+^, associated with the channel inactivation induced by the opening of the inner gate at pH 4.0.

## 4. Discussion

The conformation of the inactivated state of K^+^ channels and their modulation are still a matter of debate. In KcsA in particular, where this process was first structurally addressed, two major hypotheses have been entertained. First, it was postulated that the inactivated state of the selectivity filter has a “collapsed” conformation, similar to that seen at low K^+^ concentration, in which access to the internal S2 and S3 sites is not permitted [65]; however, a second hypothesis claimed that the filter remains in a “resting-like” conformation upon inactivation with only subtle structural changes with respect to the conductive SF [57,66,67]. Using the thermal denaturation assay, we reported previously that all the binding sites within the inactivated SF remain as accessible as in the resting state under equilibrium conditions [57], and therefore, we have no evidence to support the “collapsed” filter conformation hypothesis. Moreover, the comparison between the equilibrium constants from the binding of cations to the resting channel and to inactivated channel models indicates a decreased affinity for binding of permeant cations to the inactivated channel states, which is particularly notorious of K^+^ [57,68,69]. Electrophysiologically, although under non-equilibrium experimental conditions, inactivation has long been associated with a loss of K^+^ from the SF [70], which explains why C-type inactivation is favored at low K^+^ concentrations in eukaryotic potassium channels [19,35,71], while a higher concentration of the cation prevents it [71,72,73,74,75]. These led to postulating the “foot in the door” or the “ion depletion of the pore” hypothesis, which propose that the presence of ions inside the selectivity filter is essential to stabilize it in the conductive conformation [58,59,60]. Thus, it could be speculated that the drop in the affinity we detected in our equilibrium thermal denaturation assays could increase the probability of K^+^ loss from the filter and partly explain inactivation, although the details on the molecular mechanisms involved in the process remain to be stablished.

Another piece of evidence relevant to the work presented here comes from studies on macroscopic and single-channel current measurements of KcsA reconstituted into liposomes containing added PA [20]. It was observed that the presence of the anionic phospholipids in the liposomes caused a drastic reduction of channel inactivation. As those experiments used liposomes, it was not clear whether the lipid effects were exerted directly on the protein or through modification of bilayer properties. Nonetheless, similar effects to those of PA on KcsA currents were seen when either one of the “non-annular” arginine residues were mutated to alanine, regardless of the added anionic lipid [20]. Finally, NMR and Molecular Dynamics simulation studies concluded that, indeed, the effects of lipids were a consequence of their binding to the “non-annular” arginines (R64 and R89) in KcsA [20,22,43,54]. According to Molecular Dynamics predictions, such lipid binding prevents interaction between those arginines and the inactivation triad (E71-D80-W67) behind the selectivity filter, which seems critical to stabilize the triad and to allow inactivation.

Here, we provide experimental evidence for the first time that binding to the KcsA channel of the anionic POPA in DLMs specifically induces a change in the conformational equilibrium of the channel SF when the inner gate is in the open state. Such change consists of increasing the affinity of the channel to bind to the permeant K^+^, thus stabilizing a conductive-like channel state at physiological amounts of K^+^ and contributing to maintain ion occupancy in the pore. As we used DLMs, no lipid bilayers are formed, and therefore, the observed effects should be attributed to direct interaction of the lipid with the protein, acting as an allosteric effector. The term “specific” used before in reference to the lipid effects on the channel has several connotations: First, it modifies the binding of K^+^ but not that of Na^+^, which remains unperturbed, ruling out a merely electrostatic phenomenon of cation attraction by the extracellular channel region containing the anionic phospholipid bound at the “non-annular” sites. Second, no lipid effects were observed when the zwitterionic POPC, instead of the anionic POPA, was present in the DLMs. Third and finally, the effects of POPA on the channel protein were only observed at pH 4.0, which was the experimental condition of reference to open the channel inner gate.

From the above results on the increase in K^+^ affinity caused by POPA on the channel with an open inner gate, it is tempting to speculate that bound POPA protects the channel against inactivation; however, we cannot measure K^+^ currents and its rate of inactivation in the DLMs to provide a direct, experimental evidence in this respect. Instead, we decided to characterize the K^+^ binding behavior of two well-known non-inactivating KcsA channel mutants, E71A and R64A, and compare these with the observed effects of POPA on KcsA. Both E71A W67 KcsA and R64A W67 KcsA non-inactivating mutants in plain detergent–protein micelles (i) behaved similarly in the K^+^ titrations, (ii) were not very sensitive to changes in pH, and (iii) their *K_D_* values for K^+^ binding was in the low mM range in all cases. These quite closely emulate the behavior of W67 KcsA in the POPA DLMs at pH 4.0, strongly suggesting that, indeed, the presence of POPA bound to the channel and the increase in K^+^ affinity caused by the lipid under those conditions protects against inactivation. Moreover, such effects of bound POPA on increasing K^+^ affinity by the channel, also provides a reasonable mechanism to explain the previously reported effects of anionic phospholipids on slowing down the rates of channel inactivation [20,52]. In other words, the effects of lipids on inactivation previously reported by electrophysiological methods would be mediated by the lipid-induced changes in the affinity of the selectivity filter to bind to the potassium observed in the DLMs.

KcsA is a prokaryotic K^+^ channel with no associated human physiopathological conditions. Nonetheless, the idea that an ion channel may be modulated allosterically by binding of ligands to “non-annular” sites contributes new pharmacological targets and paves the way to the design of “lipid-like” molecules as potential channel-modulating drugs. In fact, “non-annular” sites have also been detected in some instances in eukaryotic channels, which also modulate channel function [2,4,5,48,76,77].

A corollary from the main results from above, with the W67 channel at pH 7.0 or 4.0, is that, at sufficiently high K^+^ concentration, the anisotropy values as well as the derived intersubunit distances, are almost undistinguishable between the resting and the inactivated states of the channel, at least from the data provided by the W67 reporter group used in the experiments. This seems to contribute further evidence that the inactivated channel state is “resting-like” rather than “collapsed”, as discussed previously [57].

## Data Availability

The data are contained in the article.

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
