# Peer review of "Anionic Phospholipids Shift the Conformational Equilibrium of the Selectivity Filter in the KcsA Channel to the Conductive Conformation: Predicted Consequences on Inactivation"

_biomedicines, 2023, doi:10.3390/biomedicines11051376_

Round 1

Reviewer 1 Report

This work reports the effects of anionic not zwitterionic lipids on increasing the affinity for K+ of KcsA channels in micelles, an effect suggested to be observed only at pH 4.0 when the pH-sensitive inner gate is open.

point 1. The main concern overall is that interpretations of data are oversimplified. Methods are assumed to detect only the ion interactions that occur specifically at the selectively filter. However ion binding sites with potential effects on conformational state and stability are likely to be exposed over the entire peptide, particularly since the subunits are unlikely to be assembled normally, given that "DLMs lack an organized bilayer" (line 39), and "lack a membrane bilayer structure" (lines 205-6). The underappreciated complexity of binding interactions with K+ is highlighted by the multiphasic components of dose-response curves illustrated in Fig 2. Yet, overall responses to increasing K+ at pH 7 (Fig 2A) and pH 4 (Fig 2B) appear surprisingly similar based on visual inspection, raising concerns about interpretations of different outcomes as showing the open state versus the closed state channel behaviors.

Similarly, it does not make sense that single values for dissociation constants (Table 1) can be deduced from curves that clearly show multiple components are contributing to the results.

point 2. The title implies that consequences of lipids on inactivation will be assessed, but the data show at best show channel population shifts consistent with effects of lipids and cations on general conformational properties. Nonetheless, these results are more deeply interpreted as single site effects that serve as accurate proxy measures for channel state determinations. The processes of channel inactivation in the treatment groups are not directly measured in the results shown.

Prior work by the group has used patch clamp to assess functional properties. A similar approach here would add strength to the conclusions. In particular, independent criteria are needed for the proposed allocations of outcomes into channel states (collapsed, partly empty and fully occupied conformations), which are not resolvable with the global assessment tools used here to look at populations of channels not necessarily uniform in assembly or conformation.

point 3. The authors refer to prior work [46], stating that the quadruple mutant channel used here (referred to as W67 KcsA) is structurally and functionally similar to WT KcsA, as evidence justifying its use in monitoring fluorescence anisotropy as a parallel method for monitoring changes in intrasubunit distances. An important detail overlooked here is that the authors showed previously this mutant is less thermostable than WT, a difference that is directly relevant to methods in the current MS.  

What is the evidence that the tetramer melting temperature reflects ion binding specifically at the selectivity filter, as opposed to conformation-influencing sites over the entire protein, particularly when the channel is being observed in a chaotic environment without the bilayer organisation essential for channel stability or proper assembly?

point 4. Surprisingly, data for Na+ and low K+ concentrations were dismissed by the investigators as being 'not too informative' and were omitted from the figures. Work published by the group in 2006 observed that  "concentrations of Na+ well below its dissociation constant and even in the presence of higher K+ concentrations, cause a remarkable decrease in the protein thermal stability", suggesting that low concentration effects cannot be summarily dismissed. The Na effect in Fig 1 is unlikely to be interpretable as showing only one site for candidate interactions.

point 5. The Scheme 1 Figure and legend merit revision. The Scheme 1 legend refers to DPPA (line 66), an abbreviation not defined in text or legend. Other terminology also is imprecise in the legend; for example "the most intracellular part of the C-terminal domain" (line 63) doesn't make sense, and likely the authors meant that the most distal region of the C-terminal sequence was omitted from crystal structural analysis.

The Scheme 1B image looks distorted (stretched horizontally). The positions of DPPA spheres referred to in the legend are not clearly defined; it would be good to also specify colors when used. To support the stated conclusions that  R64 and R89 interact with lipid, insets, showing hypothetical bond interactions would be useful; in the current graphic the evidence for this conclusion is not clear. For panels C and D, it is important to clarify in the legend whether the structures shown for the selectivity filter are hypothetical, or demonstrated (based on data presented in this MS), or adapted from published work (citing sources).

point 6. Extensive prior work has identified lipids in KcsA crystal structures [background citing 8,29,31-42], with functional, structural and molecular dynamics simulation studies [39,43– 108], which seems to leave the gap in knowledge being addressed in this MS unclear. The stated goal is to study effects of lipids in modulating interactions with permeating and non-permeating cations, and changes in 'conformational equilibrium' of KcsA, but these are assessed with low resolution methods (see pt 1 above), leaving the advances made by the current work uncertain.

Minor points:

Line 155: " presence of a given ligand can induce either an increase or a decrease of the observable." This sentence is incomplete; a change in the observable what?

Lines 160-1611: What is meant by "adequate amounts of NaCl or KCl as required"?

Line 225: plural is not the same as possessive (KD's)

The term "evolve"  (used in line 28) is not accurate.

Grammar errors throughout need correction.

Reviewer 2 Report

The study by Renart et alii describes the allosteric effect of the anionic phosphatidic acid species, POPA, on the function of the KcsA channel. The work is well written, the results are of interest for a wide audience and the conclusions drawn are in line with the observations reported. However, a number of minor issues should be addressed to improve the overall quality of the paper.

1. Beyond the nature of the binding (covalent or not), lipids can interact with transmembrane proteins in an indirect (via biophysical properties of the membrane) or direct manner. Thus, in line 33, where it reads "...a more direct effect..." the word more is redundant and might be deleted.

2. Please define the non-annular region in the discussion section (line 426) and the controversial nature of this region (only one or two sentences, as this is not the main topic of the paper).

3. An example of thermal analysis, including the deconvolution curves generated, would benefit the work. It might be included either in the main text or as a supplemental figure, plus its corresponding caption and text.

4. Please define WT as "wild type" the first time it is used.

5. For binding experiments, please indicate the Bmax, in addition to the Kd, if available.

6. Kd values range from 170 mM (micromolar) and 91 mM (millimolar: Table 1). Please discuss in one or two sentences the molecular nature and roles of these two binding sites (e.g., is only one physical site with two behaviors, it corresponds to two physical binding pockets, they differentially regulate the channel activity, etc.)

7. As the audience of Biomedicines are interested in the biomedical applications of cellular molecular entities, the work would increase its potential interest for the audience if, in the discussion section, the pathophysiological roles of KcsA were discussed.

8. In addition, the present results suggest that the membrane lipid composition, mainly the POPA levels, regulates the activity of the channel. Moreover, other anionic lipids (other phosphatidic acid species than palmitoyl-oleoyl PA, phosphatidylserine, phosphatidylinositol, cardiolipin, etc.) might regulate (or not) these KcsA channels. Please discuss these facts in the context of the "membrane lipid therapy" theory and how the present results extend the scope of this approach. Discuss the potential development of "biomedicines" based on the present results.

Round 2

Reviewer 1 Report

All concerns raised in the prior review have been thoughtfully and capably addressed by the authors. The revisions that have been incorporated in background, methods and interpretation of outcomes sections have substantially enhanced the clarity of the work, reaching out to a broad non-expert audience.  The only suggestion for the authors' consideration is to check that the resolution of the figures is acceptable. In the downloaded document used for this review, the graphics are slightly pixelated and blurred. (Using pdf versions of the images could be helpful for conserving image quality during the transfer into a word document template.)